# Improving Simulations of Rice in Response to Temperature and CO$_2$

Sanai Li [1,2], David H. Fleisher [1,*], Dennis Timlin [1], Jinyoung Barnaby [3], Wenguang Sun [4], Zhuangji Wang [5] and V. R. Reddy [1]

1 Adaptive Cropping Systems Laboratory, USDA-ARS, Beltsville, MD 20705, USA
2 Texas A&M AgriLife Research Center at Beaumont, 1509 Aggie Drive, Beaumont, TX 77713, USA
3 Floral and Nursery Plants Research, USDA-ARS National Arboretum, Beltsville, MD 20705, USA
4 Nebraska Water Center, University of Nebraska Lincoln, Lincoln, NE 68588, USA
5 Department of Plant Science and Landscape Architecture, University of Maryland, College Park, MD 20742, USA
* Correspondence: david.fleisher@usda.gov

**Abstract:** Crop models are frequently used to assess the impact of climate change responses. Evaluation of model performance against empirical data is crucial to establish confidence, particularly for rice (*Oryza sativa* L.), one of the world's important cereal crops. Data from soil-plant-atmosphere-research (SPAR) chambers and field plots were used to assess three versions of the ORYZA model to a range of climate conditions. The three versions were: V1–the original, V2–V1 plus a revised heat stress component, and V3–V2 plus a coupled leaf-level gas exchange algorithm. Comparison against SPAR datasets, which covered a range of temperatures at two CO$_2$ levels, indicated successive improvement in yield predictions with the model version. Root Mean Square Error (RMSE) decreased by 520 and 647 kg ha$^{-1}$ for V2 and V3, respectively, and Wilmott's index of agreement improved by 10 and 12% compared with V1 when averaged across 20 treatments and three cultivars. Similar improvements were observed from 17 field dataset simulations with two additional varieties. These results indicated the importance of improving heat sterility functions and carbon assimilation methodologies that incorporate direct responses to air temperature and CO$_2$ concentration in rice models. Accounting for cultivar differences in thermal sensitivity is also an important consideration for climate assessments.

**Keywords:** crop model; photosynthesis; high temperature; CO$_2$; spikelet fertility; heat stress

## 1. Introduction

Rice (*Oryza sativa* L.) is the main staple food for over half the world's population and is of particular importance to global food security [1,2]. The crop is vulnerable to high air temperatures, and projected climate changes are thus of great concern [3,4] given that a substantial portion of current production regions are already at, or above, thresholds associated with heat stress. While rice shows positive yield and growth responses to elevated atmospheric carbon dioxide concentration (CO$_2$) [5–7], there is evidence that the CO$_2$ fertilization effect is minimized as temperatures increase [3,8]. Explanatory crop models are frequently used to assess and identify the impact and potential adaptation strategies associated with current and future climate impacts on crop production [9–13]. However, uncertainties remain regarding the ability of rice crop models to simulate yield responses to rising CO$_2$ and temperatures [14–16]. The current study focuses on the extent to which changes to thermal stress and representation of gas exchange processes in a popular rice model resulted in improved yield and biomass predictions.

Implementation of heat responses on plant growth and development, particularly at high and low temperature extremes, is one of the largest sources of uncertainty among crop models [15,17]. This is a particular concern for rice, where even a few hours of high-temperature exposure (e.g., above 33.7 °C) can reduce pollen viability and spikelet

fertility, with a consequent reduction in grain yield [18,19]. Popular rice models such as ORYZA [20,21] incorporate the effects of a broad temperature range on developmental rates, leaf expansion, photosynthesis and respiration rates, and other growth and development components. However, predictions of high or low-temperature impacts on yield can be inaccurate [22,23] despite options for simulating high heat stress and cold sterility effects on spikelet fertility and leaf expansion rate. Similarly, a recent rice model intercomparison study [24] found that 14 crop models underestimated the negative effects of heat stress on grain yield due to a lack of full accounting for short-term extreme heat exposure during anthesis.

Simulated responses to elevated $CO_2$ are generally more consistent among crop models than temperature extremes [25], but methodologies vary considerably. This can lead to differences when projecting interactions with temperature and other factors. A recent survey of crop modeling approaches indicated that empirical adjustments to radiation use efficiency (RUE) and/or transpiration efficiency coefficients are the most frequently used [10]. While fairly robust, these and other empirical approaches can result in inaccuracies when assessing the effects of $CO_2$ along with high growth temperatures and limited water availability [26]. Predicting the response of rising $CO_2$ and temperature is complicated by the influence $CO_2$ indirectly exerts on leaf temperature due to stomatal closure and evaporative cooling [27,28]. Mimicking the photosynthetic acclimation effect [26] in models is also challenging. Methods that directly account for the influence of elevated $CO_2$ on stomatal closure and link gas exchange with an energy balance at leaf surfaces have therefore been suggested as more appropriate for climate change studies [29]. Coupled gas exchange–energy balance methodologies have been recently implemented and tested in corn, potato, and soybean models [30–32]. A related energy balance approach using sub-models for photosynthesis, stomatal conductance, and transpiration was recently incorporated into ORYZA [33] and shown to provide more accurate estimates of canopy photosynthesis under different $CO_2$ levels over the growing season as compared with the original model. However, this modification has not been fully evaluated, particularly in conjunction with varying growth temperature levels.

The objective of this study was to evaluate the effectiveness of gas exchange, and heat stress modifications implemented in the ORYZA rice model with respect to five cultivars. In this effort, we (1) initially modified the model to more accurately simulate growth duration over the broader temperature range common to U.S. conditions [23], (2) modified and parameterized the methods for simulating the impact of high heat on spikelet sterility, and (3) tested the extent to which adding the heat stress routine along with an existing coupled leaf gas exchange–energy balance model would improve predictive capability in reproducing effects of temperature and $CO_2$ on yield and above-ground biomass. Results highlight the effect these incremental improvements have on increasing the accuracy of rice model predictions for current and possible future climate impacts, which can serve as foundational pieces for improvements to other crop models.

## 2. Materials and Methods

### 2.1. Datasets

Soil-Plant-Atmosphere-Research (SPAR) growth chamber data were used to evaluate different model version responses for three cultivars grown across a wide range of growth temperatures and two $CO_2$ levels. Field data from the U.S. Mississippi Delta were used to further test the response of two additional cultivars over several years of data with different planting dates. Above-ground biomass and grain yield were obtained from multiple experiments previously conducted in SPAR chambers located at the United States Department of Agriculture—Agricultural Research Service (USDA-ARS), Beltsville, Maryland, and the University of Florida, Gainesville, Florida, U.S. The experimental data were reported in [6,34–38] and utilized IR30, Cocodrie, and Jefferson cultivars (Table S1). The IR30 cultivar studies included 17 datasets. Five temperature treatments ranged from 25/18 °C to 37/30 °C day/night thermoperiods under elevated $CO_2$ (660 ppm). Two additional treat-

ments at 28/21 °C and 31/31 °C were conducted at both ambient and elevated $CO_2$ levels, along with an additional study at 31/31 °C with six different $CO_2$ levels. Data for Cocodrie and Jefferson cultivars came from [6] and included four constant 24 h air temperature treatments of 24, 28, 32, and 36 °C at elevated $CO_2$ (700 ppm) with an additional temperature treatment of 28 °C at ambient $CO_2$. Air temperature, $CO_2$ concentration, and active photosynthetic radiation (PAR) were available on-site for Jefferson and Cocodrie cultivars. Solar radiation values were obtained at the Florida location from AgMERRA and AgCFSR datasets [39].

Experimental field data were obtained from performance trials of paddy-grown rice within the U.S. Mississippi Delta [40]. Data for two varieties, a heat stress tolerant hybrid, Clearfield CLXL753 (RiceTec, Inc., Alvin, TX, USA; https://www.ricetec.com/wp-content/uploads/2017/10/2018-Product-Characteristics_HR.pdf accessed on 20 November 2022), and a conventional tropical japonica variety, Wells, were selected based on prominence in the dataset. The data ranged from 2012 to 2015 and included multiple planting dates between March 11 and June 18 of each year for a total of 17 individual test sets. This variation in planting dates exposed the varieties to different temperatures during the flowering period, during which maximum daily temperatures varied from 27 to 35 °C. Measurements included planting and emergence dates, days to half-inch internode elongation and 50% heading, and grain yield. Weather data, including daily maximum and minimum temperature, rainfall, solar radiation, relative humidity, and wind speed, were obtained from the Integrated Agricultural Information and Management System (iAISM) [41].

*2.2. Crop Model Modifications*

Three model versions were evaluated, the default version ORYZA-V1, which included a modification for the non-linear temperature dependency of plant development; ORYZA-V2, which added a modified heat stress component; and ORYZA-V3 which added a coupled leaf gas exchange methodology along with the changes in ORYZA-V2. The original ORYZA phenology model used a bilinear temperature relationship to predict development rates (Equations (A1)–(A3)). We replaced this relationship with a normalized beta function (Equation (A11), which was previously shown to more accurately simulate the response of growth duration over a broader temperature range [42,43]. This version of the model was used as the default (ORYZA-V1). Prior to model calibration, cardinal base temperatures for early leaf expansion rates were set to 10 °C for japonica rice varieties (Cocodrie and Jefferson) and 12 °C for indica rice (IR30) to account for differences in U.S. production environment [44,45].

The original ORYZA photosynthesis routine (Equations (A4)–(A6)) used hourly leaf-level predictions for sunlit/shaded leaves at three depths in the canopy using absorbed radiation and light use efficiency values [20]. This method was previously replaced by the authors with a coupled leaf-level gas exchange model within an energy balance (Equations (A12)–(A20)) at the leaf surface [33]. In this approach, leaf net photosynthesis is simulated with the Farquhar, von Caemmerer, and Berry (FvCB) biochemical model [46], which is influenced by direct inputs for solar radiation, leaf temperature, wind speed, leaf surface $CO_2$, and relative humidity. These equations are linked with the Ball-Woodrow-Berry model for the stomatal conductance [47]. Both models are nested within an energy balance which combines the output from the stomatal conductance model to estimate leaf temperature and transpiration. These three models represent a system of interdependent equations which are solved iteratively for leaf temperature, $T_L$ (Equation (A20)). Photosynthesis and stomatal conductance are re-computed at each step until the iteration converges on a final value of $T_L$. A detailed description can be found in [33] and supporting materials (Equations (A12)–(A20)). Photosynthetic values for the FvCB model were originally obtained using SPAR chamber data for the CLXL745 rice hybrid at different developmental stages under ambient $CO_2$ (410 ppm) at a 28/23 °C day/night thermoperiod. The same parameter values were used for all cultivars in this study on the assumption that such responses were conserved across varietal lines.

The potential number of spikelets per rice plant is reduced by high or low thermal stress. The original high-temperature stress relationship in the ORYZA model was simulated as an exponential function of average daily maximum temperature (Equation (A10)) during the flowering period. Subsequent versions of this approach have been added as user-selected options by other researchers. For example, van Oort et al. [22] estimated both the hour during the day when flowering occurs ($t_{peakfl}$) and the air temperature at this time ($T_{air}(t_{peakfl})$) as in Equations (1) and (2):

$$t_{peakfl} = t_{sunrise} + 12.7 - 0.348 \times T_{min7} \tag{1}$$

$$T_{air}\left(t_{peakfl}\right) = T_{min} + (T_{max} - T_{min}) \times \sin\left(\pi \times \frac{t_{peakfl} - t_{sunrise}}{DL + 2 \times 1.5}\right) \tag{2}$$

where $t_{sunrise}$ is estimated time of sunrise calculated as the difference of 12—half of the daylength (DL) period, $T_{min7}$ is the average daily minimum temperature, $T_{min}$, of the preceding seven days, and $T_{max}$ is the daily maximum temperature.

This approach [18] does not account for potential genetic variation with respect to heat tolerance among varieties. These relationships were thus newly modified as part of the current study with a function obtained from a new statistical analysis of experimental data from [48] and parameterized to the following:

$$HEATTT = max\left(\sum_{f}^{ef}\left(T_{air}\left(t_{peakfl}\right) - T_c\right), 0\right) \tag{3}$$

$$S_h = min\left(1.0, \frac{\exp(-0.65 - H_s \times HEATTT)}{1 - \exp(-0.65 - H_s \times HEATTT)}\right) \tag{4}$$

where *HEATTT* is the cumulative heating degree-days since anthesis, $T_{air}$ is the hourly air temperature during peak flowering time $t_{peakfl}$ as defined earlier, $f$ and $ef$ are the dates of flowering and end of flowering, $T_c$ is a cultivar dependent critical temperature for heat stress during the flowering period, $H_s$ is a heat sensitive factor ranged from 0.1 to 0.25 (value of 0.167 is applied in this study) for tolerant and sensitive cultivars respectively (Figure S1), and $S_h$ is the fraction of florets which form grain.

### 2.3. Model Calibration and Evaluation

Calibration values were obtained for ORYZA-V1 (Table S2) and used for all three versions. Five varieties were calibrated, three (IR30, Jefferson, and Cocodrie) using the SPAR data and two (Wells and XL753) from field data. All phenological parameters, including development rates (Table S2), were obtained using the calibration program "pheno_opt_rice2" [49]. An independent set of SPAR data from 1988 (not shown in Table S1) with day/night temperature treatments from 28/21 to 40/33 °C was used to obtain IR30 calibration parameters. An independent set of SPAR data from [6], which covered temperature treatments from 23/19 °C to 35/31 °C, was used to develop calibration values for Cocodrie and Jefferson with the assumption that tropical japonica subspecies would have similar phenological responses. Growth parameters describing biomass partitioning and specific leaf area were fixed to those contained within the ORYZA standard crop file. Spikelet growth factor (SPGF), which sets the potential number of spikelets that can flower, was estimated from observed grain yield data under a control temperature treatment of 28/25 °C at ambient $CO_2$ from [6,34,35] for each cultivar. Critical temperatures for heat stress on spikelet fertility (Table S2) were obtained from [18,50]. Cultivars CXL753 and Wells were calibrated using 2013 field data from [40], which was also used to calibrate heat stress thresholds.

Simulations were conducted for all datasets in Table S1 to evaluate the extent to which the temperature stress and gas exchange subroutine modifications were able to improve the ability of ORYZA to estimate the response of yield to rising temperatures and $CO_2$. Simulations were subsequently conducted for the field data in Table S1. Rice was simulated without limitation of irrigation and fertilization as per literature reports

for both the controlled environment and field publications. Evaluation metrics included the ratio of simulated to observed response as a measure of relative error; root mean square error (RMSE is in Equation (A21)), and Wilmott's index of agreement (d) [51] in Equation (A22).

## 3. Results

### 3.1. Evaluation with SPAR Chamber Data

Only ORYZA-V3 and V1 above-ground biomass predictions were compared because V2 and V1 version responses were nearly identical. Results varied according to cultivar (Table 1). The ratio of simulated to observed values, an indicator of relative error, was similar between model versions for IR30 (<5% error on average), and Wilmott's index of agreement (d) was nearly identical (Table 1). However, RMSE was about 380 kg ha$^{-1}$ higher for ORYZA-V3. This error was associated with an overestimated biomass for the 1990 660 ppm 28/21 °C treatment. While V3 generally simulated a stronger biomass response with elevated $CO_2$, there was not always a consistent pattern with temperature. In contrast, RMSE was less for ORYZA-V3 by as much as 1300 kg ha$^{-1}$ for Cocodrie and 190 kg ha$^{-1}$ for Jefferson. Both model versions over-predicted aboveground biomass at the 36 °C treatments for these two cultivars, suggesting the response of vegetative growth, such as leaf area expansion, at high temperatures may be over-emphasized.

**Table 1.** Comparison of observed (Obs) and simulated (Sim) above-ground rice biomass for ORYZA-V1 and ORYZA-V3 model versions in response to $CO_2$ and day/night temperature (*T*) treatments for three rice varieties. The ratio of simulated to observed final biomass (Sim/Obs) was indicated for each individual treatment, and Wilmott's index of agreement (d) and RMSE computed across all treatments for each variety.

| Variety | Year | CO$_2$ (ppm) | *T* (°C) | Obs (kg ha$^{-1}$) | ORYZA-V1 | | ORYZA-V3 | |
|---|---|---|---|---|---|---|---|---|
| | | | | | Sim (kg ha$^{-1}$) | Sim/Obs | Sim (kg ha$^{-1}$) | Sim/Obs |
| IR30 | 1987 | 330 | 31/31 | 12,925 | 15,078 | 1.17 | 14,540 | 1.12 |
| | 1987 | 330 | 31/31 | 16,920 | 14,678 | 0.87 | 14,790 | 0.87 |
| | 1987 | 660 | 31/31 | 17,625 | 20,856 | 1.18 | 22,546 | 1.28 |
| | 1987 | 660 | 31/31 | 21,855 | 19,828 | 0.91 | 22,140 | 1.01 |
| | 1989 | 330 | 28/21 | 15,275 | 16,038 | 1.05 | 15,999 | 1.05 |
| | 1989 | 660 | 25/18 | 18,330 | 22,005 | 1.20 | 21,369 | 1.17 |
| | 1989 | 660 | 28/21 | 20,915 | 20,811 | 1.00 | 22,215 | 1.06 |
| | 1989 | 660 | 34/27 | 19,035 | 16,794 | 0.88 | 18,074 | 0.95 |
| | 1989 | 660 | 37/30 | 16,685 | 13,607 | 0.82 | 13,265 | 0.80 |
| | 1990 | 330 | 28/21 | 17,836 | 19,032 | 1.07 | 19,472 | 1.09 |
| | 1990 | 660 | 28/21 | 22,869 | 23,614 | 1.03 | 27,016 | 1.18 |
| | Average | | | 18,206 | 18,395 | 1.02 | 19,221 | 1.05 |
| | d | | | - | 0.84 | - | 0.85 | - |
| | RMSE (kg ha$^{-1}$) | | | - | 2232 | - | 2618 | - |
| Cocodrie | 2000 | 350 | 28/28 | 18,036 | 15,174 | 0.84 | 15,979 | 0.89 |
| | 2000 | 700 | 24/24 | 24,336 | 20,558 | 0.84 | 20,603 | 0.85 |
| | 2000 | 700 | 28/28 | 25,236 | 19,200 | 0.76 | 21,664 | 0.86 |
| | 2000 | 700 | 32/32 | 24,048 | 17,918 | 0.75 | 20,643 | 0.86 |
| | 2000 | 700 | 36/36 | 12,312 | 14,267 | 1.16 | 14,945 | 1.21 |
| | Average | | | 20,794 | 17,423 | 0.87 | 18,767 | 0.93 |
| | d | | | - | 0.74 | - | 0.84 | - |
| | RMSE (kg ha$^{-1}$) | | | - | 4479 | - | 3145 | - |

**Table 1.** *Cont.*

| Variety | Year | CO$_2$ (ppm) | $T$ (°C) | Obs (kg ha$^{-1}$) | ORYZA-V1 | | ORYZA-V3 | |
|---|---|---|---|---|---|---|---|---|
| | | | | | Sim (kg ha$^{-1}$) | Sim/Obs | Sim (kg ha$^{-1}$) | Sim/Obs |
| | 2000 | 350 | 28/28 | 15,876 | 15,224 | 0.96 | 16,033 | 1.01 |
| | 2000 | 700 | 24/24 | 25,848 | 20,695 | 0.80 | 20,670 | 0.80 |
| | 2000 | 700 | 28/28 | 21,384 | 19,329 | 0.90 | 21,738 | 1.02 |
| Jefferson | 2000 | 700 | 32/32 | 20,772 | 18,005 | 0.87 | 20,718 | 1.00 |
| | 2000 | 700 | 36/36 | 10,152 | 14,271 | 1.41 | 14,941 | 1.47 |
| | Average | | | 18,806 | 17,505 | 0.99 | 18,820 | 1.06 |
| | d | | | - | 0.82 | - | 0.84 | - |
| | RMSE (kg ha$^{-1}$) | | | - | 3341 | - | 3159 | - |

The ORYZA-V1 version accurately simulated the IR-30 cultivar yield response to temperature and CO$_2$ when maximum temperatures were below 32 °C. However, yields were over-estimated above 32 °C by as much as 87% (Table 2). Performance was identical with ORYZA-V2 for all data sets at the lower temperature ranges. However, the observed declines in grain yield above 32 °C were more accurately simulated with about 6% relative error. Yields for most other temperature treatments and cultivars were under-predicted by both model versions. The ORYZA-V3 model replicated grain yield more accurately than either -V2 or -V1 model versions. There was a 10% or larger increase in the agreement index compared with -V1 and a lower RMSE between 400 to about 800 kg ha$^{-1}$. There was little difference between ORYZA-V2 and -V3 yield predictions for Cocodrie, but modest improvements in RMSE between 140 to 270 kg ha$^{-1}$ were observed for both IR30 and Jefferson. As with above-ground biomass simulations, no consistent pattern was observed with CO$_2$ and temperature in terms of model accuracy (for temperature treatments below 32 °C). However, ORYZA-V3 tended to predict higher yield values than -V2 or -V1 versions, especially for Cocodrie and Jefferson.

Yield sensitivity to the temperature at two CO$_2$ levels is illustrated in Figure 1. All model versions tracked the differences in temperature response for cultivars below 34, 36, and 37 °C. At these higher temperatures, substantial over-predictions from ORYZA-V1 were evident. All model versions also showed a positive response to CO$_2$ enrichment, demonstrated by the observed values. However, ORYZA-V3 exhibited more accurate changes to elevated CO$_2$ at a given temperature treatment. For example, there were paired CO$_2$ treatments at 28 °C and 31 °C for IR30 (Figure 1a), the results of which showed smaller relative errors than ORYZA-V2 or -V1 (Table 2). A similar response was observed among model versions for Cocodrie and Jefferson with a paired CO$_2$ treatment at 28 °C, in which ORYZA-V3 also showed reduced relative error for the elevated CO$_2$ response. Overall, simulations from ORYZA-V3 were closer to the observed yield than ORYZA-V2 for all varieties (Figure 1, Table 2).

Data from one of the 1987 IR30 studies [52], which included two planting dates (Table S1), were used to evaluate simulated CO$_2$ responses from 160 to 900 ppm at 31/31 °C (Figure 2). ORYZA-V1 and ORYZA-V3 were compared for this case because V2 and V1 predictions were identical at this temperature level. ORYZA-V3 more accurately captured the response of yield to the broader range of CO$_2$ values. Relative errors were between −7% and 9% across the 330 to 900 ppm range for ORYZA-V3 compared with under-estimates of −8 to −20% for ORYZA-V1 (Table 3). Errors were greater for the ORYZA-V3 model, however, at sub-ambient CO$_2$ levels (<250 ppm). Across all CO$_2$ levels, the index of agreement was higher (0.94 versus 0.91) and RMSE lower (872 versus 1065 kg ha$^{-1}$) for ORYZA-V3 versus ORYZA-V1, respectively. This suggested that the leaf-coupled gas exchange methodology was more accurate for projecting yields under most CO$_2$ levels.

**Table 2.** Comparison of observed (Obs) and simulated (Sim) rice grain yield for ORYZA-V1, -V2, and -V3 model versions in response to $CO_2$ and day/night temperature (*T*) treatments for three rice varieties. The ratio of simulated to observed final biomass (Sim/Obs) was indicated for each individual treatment, and Wilmott's index of agreement (d) and RMSE computed across all treatments for each variety.

| Variety | Year | CO₂ (ppm) | *T* (°C) | Obs Yield (kg ha⁻¹) | ORYZA-V1 | | ORYZA-V2 | | ORYZA-V3 | |
|---|---|---|---|---|---|---|---|---|---|---|
| | | | | | Sim (kg ha⁻¹) | Sim/Obs | Sim (kg ha⁻¹) | Sim/Obs | Sim (kg ha⁻¹) | Sim/Obs |
| IR30 | 1987 | 330 | 31/31 | 5200 | 3996 | 0.77 | 3996 | 0.77 | 4169 | 0.80 |
| | 1987 | 330 | 31/31 | 4300 | 4744 | 1.10 | 4744 | 1.10 | 5112 | 1.19 |
| | 1987 | 660 | 31/31 | 6800 | 5587 | 0.82 | 5587 | 0.82 | 6428 | 0.95 |
| | 1987 | 660 | 31/31 | 6400 | 6532 | 1.02 | 6532 | 1.02 | 7645 | 1.19 |
| | 1989 | 330 | 28/21 | 6600 | 5332 | 0.81 | 5332 | 0.81 | 5687 | 0.86 |
| | 1989 | 660 | 25/18 | 8400 | 7170 | 0.85 | 7170 | 0.85 | 7072 | 0.84 |
| | 1989 | 660 | 28/21 | 10,400 | 7294 | 0.70 | 7294 | 0.70 | 8063 | 0.78 |
| | 1989 | 660 | 34/27 | 3400 | 5726 | 1.68 | 3223 | 0.95 | 3627 | 1.07 |
| | 1989 | 660 | 37/30 | 1000 | 1867 | 1.87 | 938 | 0.94 | 995 | 0.99 |
| | 1990 | 330 | 28/21 | 8000 | 6617 | 0.83 | 6617 | 0.83 | 7077 | 0.88 |
| | 1990 | 660 | 28/21 | 10,100 | 8788 | 0.87 | 8788 | 0.87 | 9922 | 0.98 |
| | Average | | | 6418 | 5787 | 1.03 | 5475 | 0.88 | 5982 | 0.96 |
| | d | | | - | 0.88 | - | 0.92 | - | 0.96 | - |
| | RMSE (kg ha⁻¹) | | | - | 1529 | - | 1334 | - | 1062 | - |
| Cocodrie | 2000 | 350 | 28/28 | 5230 | 4958 | 0.95 | 4958 | 0.95 | 5740 | 1.10 |
| | 2000 | 700 | 24/24 | 6814 | 7457 | 1.03 | 7457 | 1.03 | 8013 | 1.11 |
| | 2000 | 700 | 28/28 | 7823 | 6331 | 0.83 | 6331 | 0.83 | 7649 | 1.00 |
| | 2000 | 700 | 32/32 | 6733 | 5873 | 0.84 | 5873 | 0.84 | 7433 | 1.06 |
| | 2000 | 700 | 36/36 | 0 | 3344 | | 984 | - | 1104 | - |
| | Average | | | 5320 | 5593 | 0.91 | 5121 | 0.91 | 5988 | 1.07 |
| | d | | | - | 0.82 | - | 0.96 | - | 0.98 | - |
| | RMSE (kg ha⁻¹) | | | - | 1446 | - | 795 | - | 829 | - |
| Jefferson | 2000 | 350 | 28/28 | 4608 | 4546 | 0.99 | 4546 | 0.99 | 5261 | 1.14 |
| | 2000 | 700 | 24/24 | 7236 | 6884 | 0.95 | 6884 | 0.95 | 7510 | 1.04 |
| | 2000 | 700 | 28/28 | 7236 | 5804 | 0.80 | 5804 | 0.80 | 7011 | 0.97 |
| | 2000 | 700 | 32/32 | 5976 | 5391 | 0.90 | 5391 | 0.90 | 6771 | 1.13 |
| | 2000 | 700 | 36/36 | 0 | 3052 | - | 934 | - | 1066 | - |
| | Average | | | 5011 | 5135 | 0.91 | 4712 | 0.91 | 5524 | 1.07 |
| | d | | | - | 0.85 | - | 0.97 | - | 0.98 | - |
| | RMSE (kg ha⁻¹) | | | - | 1539 | - | 824 | - | 681 | - |

**Table 3.** Comparison of observed (Obs) and simulated (Sim) rice grain yield for ORYZA-V1 and ORYZA-V3 model versions in response to varying $CO_2$ concentrations. Data were averaged from two different experiments, Exp I and II, that varied in planting date (22 January or 23 June) from [52]. The ratio of simulated to observed yield (Sim/Obs) was indicated along with Wilmott's index of agreement (d) and RMSE.

| Exp | CO₂ (ppm) | Observed (kg ha⁻¹) | ORYZA-V1 | | ORYZA-V3 | |
|---|---|---|---|---|---|---|
| | | | Sim (kg ha⁻¹) | Sim/Obs | Sim (kg ha⁻¹) | Sim/Obs |
| I & II | 160 | 3400 | 2386 ± 430 | 0.70 | 1547 ± 615 | 0.45 |
| I & II | 250 | 4100 | 3629 ± 837 | 0.89 | 3564 ± 647 | 0.87 |
| I & II | 330 | 4800 | 4426 ± 948 | 0.92 | 4840 ± 608 | 1.01 |
| II only | 500 | 6830 | 5435 ± 1000 | 0.80 | 6382 ± 520 | 0.93 |
| I & II | 660 | 6600 | 5966 ± 1038 | 0.90 | 7162 ± 535 | 1.09 |
| I & II | 900 | 7300 | 6390 ± 1064 | 0.88 | 7865 ± 563 | 1.08 |
| d | | - | 0.91 | - | 0.94 | - |
| RMSE (kg ha⁻¹) | | - | 1065 | | 872 | - |

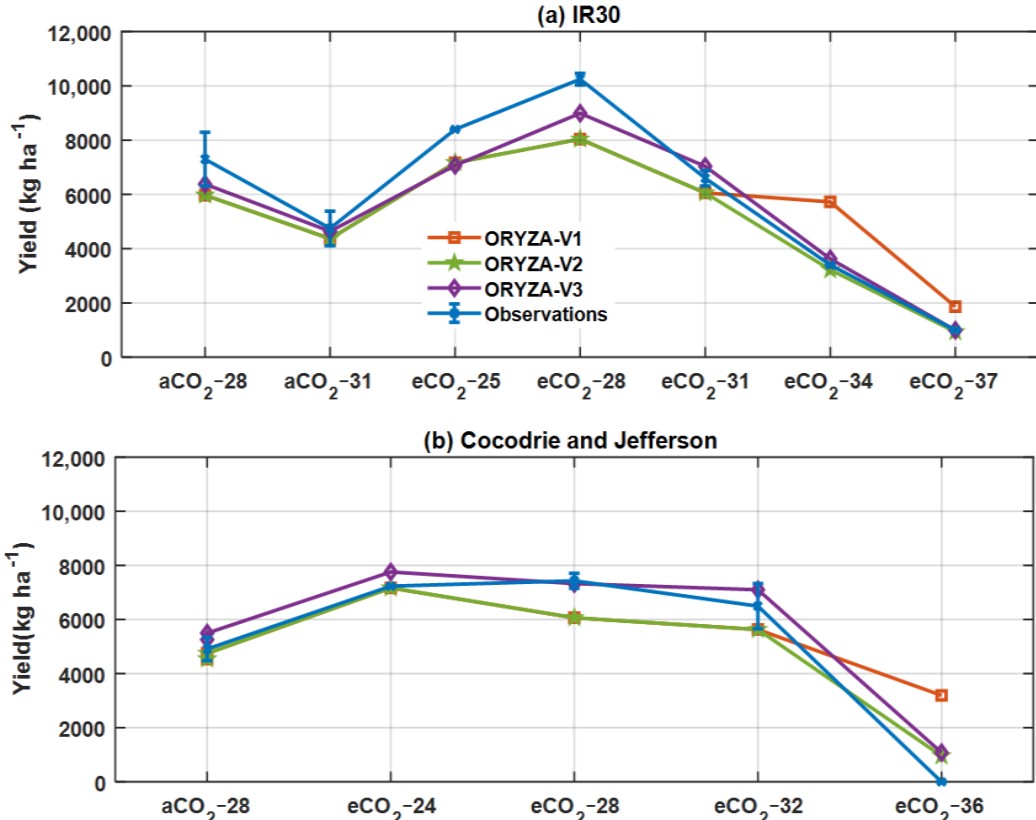

**Figure 1.** Observed and simulated response of grain yield from three model versions versus daily temperature value under ambient (aCO$_2$) or elevated (eCO$_2$) CO$_2$ for cultivars (**a**) IR30 and (**b**) Cocodrie and Jefferson using data from Table 2. Observed and simulated means for Cocodrie and Jefferson were shown as averages across varieties due to very small yield differences.

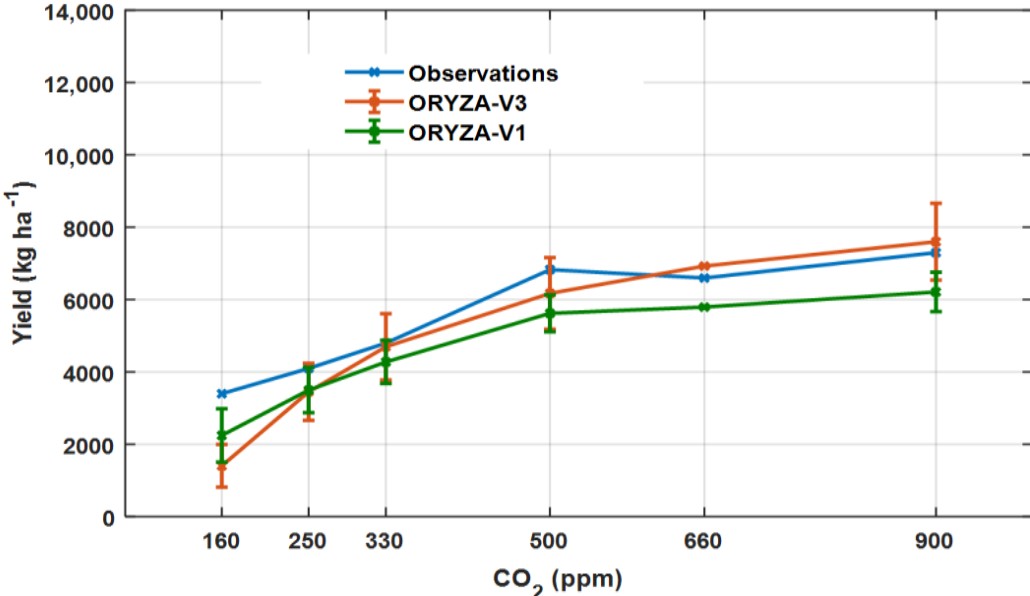

**Figure 2.** Comparison of observed (from [52]) and simulated rice yield for ORYZA-V1 and ORYZA-V3 under varying CO$_2$ concentration at constant day/night temperature of 31 °C averaged over two planting dates for the IR30 cultivar.

### 3.2. Evaluation with Field Data

Models were evaluated for Wells (Table 4) and CLXL753 (Table 5) cultivars using four years of field data, each with multiple planting dates. Calibration results for the four planting dates in 2013 show better metrics for ORYZA-V1 and -V2 than V3. This was due to the calibration values for -V1 being used for all model versions. However, evaluation data sets showed the opposite trend, where simulated yields were more accurate with increasing ORYZA versions. For example, RMSE decreased from a high of 2089 kg ha$^{-1}$ for ORYZA-V1 to a low of 1530 kg ha$^{-1}$ for -V3 for the Wells cultivar and an even larger reduction from 2429 to 955 kg ha$^{-1}$ for CLXL753. The improvements in simulation for the Wells cultivar were primarily associated with dates in 2012 and 2014, where maximum temperature values during the anthesis period were 31 °C or lower (Table 4). Given that these values were below the critical temperature threshold for heat sterility stress of 33.7 °C (Table S2), results for versions -V1 and -V2 were similar. However, the largest simulation improvements were associated with V3. For four of the treatments (5 June 2014, 3 April 2015, 21 April 2015, and 3 June 2015), ORYZA-V3 over-predicted yields compared with other versions for reasons which did not appear temperature related. Similar results were observed for CLXL753, except that ORYZA-V3 generally showed less error for 10 of the 13 sets (Table 5).

**Table 4.** Comparison of observed (Obs) and simulated (Sim) rice yield for the Wells cultivar from three versions of ORYZA from field data at Stuttgart, Arkansas, under varying planting dates from 2012 to 2015. Calibration (Cal) results for 2013 were shown (with values in Table S2) along with evaluation (Val) results for 2012, 2014, and 2015. The ratio of simulated to observed yield (Sim/Obs) was indicated along with Wilmott's index of agreement and RMSE. Average maximum daily air temperature during flowering periods was shown as $T_{max}$.

| | | | $T_{max}$ (°C) | Obs Yield (kg ha$^{-1}$) | ORYZA-V1 | | ORYZA-V2 | | ORYZA-V3 | |
|---|---|---|---|---|---|---|---|---|---|---|
| | | | | | Sim (kg ha$^{-1}$) | Sim/Obs | Sim (kg ha$^{-1}$) | Sim/Obs | Sim (kg ha$^{-1}$) | Sim/Obs |
| Cal | 28-March | 2013 | 30.2 | 9382 | 9830 | 1.05 | 10,363 | 1.11 | 11,495 | 1.23 |
| | 16-April | 2013 | 30.6 | 10,289 | 10,002 | 0.97 | 10,002 | 0.97 | 11,300 | 1.10 |
| | 30-May | 2013 | 34.2 | 8928 | 8867 | 0.99 | 8990 | 1.01 | 10,253 | 1.15 |
| | 17-June | 2013 | 29.1 | 6607 | 7754 | 1.17 | 5001 | 0.76 | 5662 | 0.86 |
| | Average | | 31.0 | 8802 | 9113 | 1.05 | 8589 | 0.96 | 9678 | 1.08 |
| | d | | | - | 0.92 | - | 0.92 | - | 0.86 | - |
| | RMSE (kg ha$^{-1}$) | | | - | 633 | - | 952 | - | 1426 | - |
| Val | 30-March | 2012 | 30.1 | 11,903 | 8415 | 0.71 | 10,663 | 0.90 | 11,863 | 1.00 |
| | 11-May | 2012 | 31.0 | 10,794 | 6664 | 0.62 | 7793 | 0.72 | 8867 | 0.82 |
| | 26-March | 2014 | 28.3 | 12,055 | 8967 | 0.74 | 8967 | 0.74 | 9995 | 0.83 |
| | 18-April | 2014 | 29.2 | 11,349 | 8818 | 0.78 | 8818 | 0.78 | 10,059 | 0.89 |
| | 2-May | 2014 | 31.1 | 9432 | 9190 | 0.97 | 9190 | 0.97 | 10,500 | 1.11 |
| | 21-May | 2014 | 33.3 | 9028 | 9216 | 1.02 | 9216 | 1.02 | 10,545 | 1.17 |
| | 5-June | 2014 | 30.4 | 6708 | 7781 | 1.16 | 7781 | 1.16 | 8953 | 1.33 |
| | 18-June | 2014 | 27.2 | 7969 | 6679 | 0.84 | 6679 | 0.84 | 7705 | 0.97 |
| | 3-April | 2015 | 33.8 | 8020 | 8432 | 1.05 | 8636 | 1.08 | 9777 | 1.22 |
| | 21-April | 2015 | 34.6 | 7011 | 6852 | 0.98 | 7930 | 1.13 | 9083 | 1.30 |
| | 5-May | 2015 | 34.7 | 8574 | 6144 | 0.72 | 7531 | 0.88 | 8650 | 1.01 |
| | 19-May | 2015 | 32.3 | 9028 | 7776 | 0.86 | 8120 | 0.90 | 9277 | 1.03 |
| | 3-June | 2015 | 28.3 | 7364 | 8368 | 1.14 | 8368 | 1.14 | 9554 | 1.30 |
| | Average | | 31.2 | 9172 | 7946 | 0.89 | 8438 | 0.94 | 9602 | 1.07 |
| | d | | | - | 0.53 | - | 0.66 | - | 0.66 | - |
| | RMSE (kg ha$^{-1}$) | | | - | 2089 | - | 1606 | - | 1530 | - |

**Table 5.** Comparison of observed (Obs) and simulated (Sim) rice yield for the CLXL753 cultivar from three versions of ORYZA from field data at Stuttgart, Arkansas under varying planting dates from 2012 to 2015. Calibration (Cal) results for 2013 were shown (with values in Table S2) along with evaluation (Val) results for 2012, 2014, and 2015. The ratio of simulated to observed yield (Sim/Obs) was indicated along with Wilmott's index of agreement and RMSE. Average maximum daily air temperature during flowering periods was shown as $T_{max}$.

| | | | $T_{max}$ (°C) | Obs Yield (kg ha$^{-1}$) | ORYZA-V1 | | ORYZA-V2 | | ORYZA-V3 | |
|---|---|---|---|---|---|---|---|---|---|---|
| | | | | | Sim (kg ha$^{-1}$) | Sim/Obs | Sim (kg ha$^{-1}$) | Sim/Obs | Sim (kg ha$^{-1}$) | Sim/Obs |
| Cal | 28-March | 2013 | 32.6 | 11,954 | 11,987 | 1.00 | 12,164 | 1.02 | 13,287 | 1.11 |
| | 16-April | 2013 | 30.0 | 12,004 | 11,969 | 1.00 | 11,969 | 1.00 | 12,773 | 1.06 |
| | 30-May | 2013 | 33.8 | 10,542 | 10,606 | 1.01 | 10,606 | 1.01 | 11,370 | 1.08 |
| | 17-June | 2013 | 31.5 | 9180 | 9442 | 1.03 | 8320 | 0.91 | 9427 | 1.03 |
| | Average | | 32.0 | 10,920 | 11,001 | 1.01 | 10,765 | 0.98 | 11,714 | 1.07 |
| | d | | | - | 0.99 | - | 0.97 | - | 0.90 | - |
| | RMSE (kg ha$^{-1}$) | | | - | 137 | - | 444 | - | 883 | - |
| Val | 30-March | 2012 | 31.3 | 13,820 | 10,265 | 0.74 | 11,849 | 0.86 | 12,330 | 0.89 |
| | 11-May | 2012 | 34.0 | 12,660 | 8146 | 0.64 | 9846 | 0.78 | 11,201 | 0.88 |
| | 26-March | 2014 | 27.6 | 11,147 | 10,936 | 0.98 | 10,936 | 0.98 | 12,188 | 1.09 |
| | 18-April | 2014 | 30.2 | 12,761 | 10,748 | 0.84 | 10,748 | 0.84 | 12,252 | 0.96 |
| | 2-May | 2014 | 31.0 | 12,105 | 10,435 | 0.86 | 10,435 | 0.86 | 11,403 | 0.94 |
| | 21-May | 2014 | 33.3 | 12,105 | 10,203 | 0.84 | 10,203 | 0.84 | 10,960 | 0.91 |
| | 5-June | 2014 | 30.6 | 9533 | 9285 | 0.97 | 9285 | 0.97 | 10,082 | 1.06 |
| | 18-June | 2014 | 27.6 | 9079 | 8142 | 0.90 | 8142 | 0.90 | 9395 | 1.03 |
| | 3-April | 2015 | 32.5 | 11,349 | 10,283 | 0.91 | 10,524 | 0.93 | 11,742 | 1.03 |
| | 21-April | 2015 | 34.6 | 11,449 | 8365 | 0.73 | 9658 | 0.84 | 11,082 | 0.97 |
| | 5-May | 2015 | 34.7 | 11,702 | 7485 | 0.64 | 9182 | 0.78 | 10,535 | 0.90 |
| | 19-May | 2015 | 32.9 | 11,349 | 9486 | 0.84 | 9896 | 0.87 | 11,316 | 1.00 |
| | 3-June | 2015 | 28.3 | 9634 | 10,200 | 1.06 | 10,200 | 1.06 | 11,150 | 1.16 |
| | Average | | 31.2 | 11,438 | 9537 | 0.83 | 10,069 | 0.88 | 11,203 | 0.98 |
| | d | | | - | 0.18 | - | 0.54 | - | 0.77 | - |
| | RMSE (kg ha$^{-1}$) | | | - | 2429 | - | 1662 | - | 955 | - |

## 4. Discussion

Metrics for above-ground biomass and yield generally improved for ORYZA-V2 and ORYZA-V3 versus ORYZA-V1 across most temperatures and $CO_2$ concentrations. The largest improvements, particularly for RMSE, occurred between V1 and V2, with incrementally fewer improvements between V2 and V3. This indicates the importance of using an improved heat stress algorithm for both current and future climate assessments on rice productivity. For example, ORYZA-V2, with the heat stress modification, resulted in an averaged RMSE reduction of 520 kg ha$^{-1}$ versus the V1 model for grain yield averaged across all SPAR data (Table 2). An average reduction of 499 kg ha$^{-1}$ was also observed across all field data (Tables 4 and 5). Improvements in RMSE for ORYZA-V3, which included the heat stress modification plus the leaf-level coupled gas exchange method, were an additional 127 kg ha$^{-1}$ and 240 kg ha$^{-1}$ for SPAR and field data, respectively. The simulation of grain yield in ORYZA is primarily driven by potential spikelet number and less by the availability of assimilate, at least over the range of conditions typically used for evaluating the rice model [20]. The improvement for the ORYZA-V2 version versus V1 observed for the field data also indicates that the occurrence of heat stress during anthesis is a characteristic of the current climate in the U.S. Mississippi Delta region. Thus, improving the simulation of heat stress on spikelet fertility and potential grain number may be more important than the methodology used for carbon assimilation alone, especially if episodic high heat events during anthesis and/or grain-fill periods continue to be problematic in rice growing regions [53].

Experimental data was insufficient to thoroughly evaluate interactions with $CO_2$ and temperature. However, the modifications implemented in ORYZA-V3 were effective in improving performance metrics for grain yield in about 80% of all treatments studied. These ranged from daytime temperatures between 24 and 36 °C and $CO_2$ levels from 330 through 770 ppm. This range of improvement also included field data in which $CO_2$ levels were at ambient levels, indicating that the gas exchange methodology is effective for today's climate as well as the future. Few other studies have directly tested the effectiveness of different methods for simulating gas exchange processes for rice in response to climate factors. Differences among yield predictions among multiple rice models to changes in temperature and $CO_2$ were observed by Li et al. [15] but were not attributed to specific carbon assimilation methods. In contrast, Hasegawa et al. [54] reported greater responsiveness to $CO_2$ for models that used a biochemical approach for modeling photosynthesis, such as that used in the ORYZA-V3. Previous work with this approach showed how the methodology could simulate reduced stomatal conductance in response to rising $CO_2$ concentration. This, in turn, decreases transpiration, increases leaf and canopy temperature (as much as 0.5 °C, not shown), and exerts an additional influence on photosynthesis [33]. However, despite these improvements, the ORYZA-V3 model over-estimated vegetative biomass for some SPAR datasets (Table 1), which may reflect the need to also account for photosynthetic acclimation [26], substrate-induced feedback inhibition [55,56], and/or the influence of nitrogen and light attenuation in the canopy on photosynthetic properties [31].

All model versions were able to capture differences in cultivar sensitivity to temperature. This includes the relative broad temperature response of IR30 in which grain yield formed as high as 37 °C (Table 2, Figure 1). Conversely, no yields were observed at 36 °C above for Cocodrie and Jefferson. ORYZA-V2 and V3 were consistently more accurate than the default version in terms of replicating these different temperature responses. Given that these varieties were from different rice sub-populations (e.g., IR30 was from indica and Jefferson and Cocodrie from tropical japonica), this result highlighted the need to consider heat sensitivity differences among cultivars (Table S2). The broad range of planting dates in field data resulted in the exposure of Wells and CLXL753 cultivars to maximum average daily temperatures during the flowering period ranging from 27.6 to 34.7 °C (Tables 4 and 5). The ORYZA-V3 and -V2 models showed significant improvement compared with ORYZA-V1 in mimicking these yields. The ORYZA-V1 used a daily maximum temperature [20] as the threshold for spikelet fertility which likely resulted in an overestimation of the impact of heat stress for several field treatments for both varieties. The rice flowers often open in the morning hours [57,58] when the air temperature is cooler than the daily maximum. Thus, the improved heat sterility model used in this study, which estimated the temperature at flowering time, was shown to predict heat-stress yield reduction more accurately under both controlled environments, where square wave temperature profiles were used (Table 2), as well as more natural conditions in which temperatures varied diurnally (Tables 4 and 5).

A strong linear relationship is known to exist between the quantity of dry matter produced during the growth period between panicle initiation through anthesis and to spikelet and grain number [20,59]. Factors associated with this response are likely related to temperature, as a driving factor of developmental rate, and solar radiation. Comparisons between simulated yields and growth stage durations for field-grown cultivars CLXL753 and Wells show high linear correlations (Table S3). Similarly, total seasonal solar radiation accumulation was also positively correlated with yields. ORYZA-V3 and V2 exhibited stronger correlations than -V1, suggesting that both the heat sterility improvement and gas exchange methodologies are important in terms of capturing expected agronomic relationships. Correlations were not significant between simulated yields and growth duration or solar radiation for the IR30, Cocodrie, or Jefferson cultivars (Table S3). This lower response may be associated with the narrower range of diurnal temperatures and the similar quantity of solar radiation available across most of the SPAR chamber treatments.

All model versions over-estimated aboveground biomass at air temperatures above 36 °C (Table 1). This was likely due to over-predictions of leaf area index as reported by van Oort et al. [22], who suggested modelers consider reduced assimilate partitioning to the leaves at lower or higher temperatures. Nagai and Makino [60] also observed differences in the ratio of whole plant leaf area to dry weight when grown under different temperatures. This suggests that specific leaf area values may need to be modified based on growth temperature to properly account for leaf area expansion and associated carbohydrate requirements. Differences in growth stage thermal sensitivities will be particularly important to account for in terms of evaluating the effectiveness of changing planting dates as a heat stress avoidance measure [61]. Further investigation into the possibility of different temperature stress thresholds associated with different rice developmental stages, as suggested by Samejima et al. [62], may also be warranted.

## 5. Conclusions

Improvements to heat sterility modules in rice models are crucial, and differences in varietal temperature threshold responses need to be accounted for. Incorporation of the simulation of gas exchange rates using a coupled leaf-level approach within a leaf energy balance substantially improved end-of-season biomass and improved grain yield simulations in both growth chamber and field studies across five different cultivars for two different rice subpopulations. These methods more accurately replicated responses compared with the original model approaches over a broad temperature range at two $CO_2$ levels and thus can increase confidence when applying this modified ORYZA model for climate assessment work. The improved ORYZA model was also able to capture the average yield response of multiple cultivars to shifting planting dates across several years of field data in which cultivars were thus exposed to different thermal regimes during flowering. Results highlight the effect these improvements have on increasing the accuracy of rice model predictions to current and future climate impacts, which can serve as foundational pieces for improvements to other crop models. Our modifications to the ORYZA model can be easily adapted to other models without substantially affecting the model structure.

**Supplementary Materials:** The following supporting information can be downloaded at: https://www.mdpi.com/article/10.3390/agronomy12122927/s1, Figure S1: The measured response of spikelet fertility (%) to hourly temperature during anthesis for heat tolerant, moderate and susceptible cultivars as measured from Matsui et al. (2001), and (b) the modeled relationship between spikelet fertility ($S_h$, 0-1) to accumulated heat stress for heat tolerant and sensitive cultivars as simulated with main text equations (Equations (3) and (4)); Table S1: Data sources and experiments conducted in SPAR units used to evaluate different model versions. Experiments included three cultivars subjected to treatments with different day/night air temperature (T), $CO_2$ concentration, and/or paddy water temperatures. Some experiments were repeated with separate planting dates as indicated; Table S2: Calibration values for cultivars IR30, Cocodrie, Jefferson, Wells, and CLXL753. Values for IR30, Cocodrie, and Jefferson were obtained from the 1988 experiment at University of Florida, and Cocodrie from the 28 and 24°C ambient $CO_2$ treatments conducted at USDA-ARS (Table 1); Table S3: Correlation coefficients from linear regression between simulated yield versus cumulative growing season solar radiation and for different phenological stage durations for the five cultivars.

**Author Contributions:** Conceptualization, D.H.F. and S.L.; methodology, S.L.; validation, S.L., W.S. and Z.W.; formal analysis, D.H.F., S.L., D.T. and J.B.; resources, V.R.R.; writing—original draft preparation, S.L. and D.H.F.; writing—review and editing, D.T., J.B. and V.R.R. All authors have read and agreed to the published version of the manuscript.

**Funding:** This research received no external funding.

**Data Availability Statement:** The experimental data presented in this study are available from the citations associated with each data source as indicated in the main text, supplemental information, and the Appendix A. Simulated data are all presented in this study as indicated. Novel computer code changes are provided as equation numbers through the main text and Appendix A.

**Acknowledgments:** This work was supported by the United States Department of Agriculture—Agricultural Research Service (USDA-ARS) CRIS-8042-11660-001-00D. Mention of a trademark or proprietary product does not constitute a guarantee or warranty of the product by the US Department of Agriculture and does not imply its approval to the exclusion of other products that also can be suitable. USDA is an equal opportunity provider and employer. All experiments complied with the current laws of the United States, the country in which they were performed.

**Conflicts of Interest:** The authors declare no conflict of interest.

## Abbreviations

| | |
|---|---|
| $CO_2$ | atmospheric carbon dioxide concentration |
| d | Wilmott's index of agreement; |
| LAI | leaf area index |
| PAR | photosynthetic active radiation |
| RMSE | root mean square error |
| RUE | radiation use efficiency |
| SPAR | soil-plant-atmosphere-research growth chambers |
| SPGF | spikelet growth factor |

## Appendix A

*Appendix A.1. Original ORYZA Model*

Appendix A.1.1. Development

ORYZA [20] integrates growth and developmental rates at a daily time step, but hourly climate data are estimated for certain functions. Development rate has a bilinear response to temperature over hourly time periods. Hourly temperatures ($T_h$, °C) are estimated by a sinusoidal function of daily minimum ($T_{min}$, °C) and maximum ($T_{max}$, °C) air temperatures:

$$T_h = \frac{(T_{max} + T_{min})}{2} + \frac{((T_{max} - T_{min}) \times \cos(0.2618(h - 14))}{2} \tag{A1}$$

where $h$ is the time of day. Hourly increments in heat units ($HUH$, °Cd h$^{-1}$) are given by Equation (A2):

$$HUH = \begin{cases} 0 & T_h \leq T_{base} \; or \; T_h \geq T_{high} \\ \frac{(T_h - T_{base})}{24} & T_{base} \leq T_h \leq T_{opt} \\ (T_{opt} - \frac{(T_h - T_{opt}) \times (T_{opt} - T_{base})}{(T_{high} - T_{opt})})/24 & T_{opt} \leq T_h \leq T_{high} \end{cases} \tag{A2}$$

where $T_{base}$, $T_{opt}$, and $T_{high}$ are the base, optimal, and maximum temperature for development. The daily increment in heat units (HU, °Cd d$^{-1}$) is then calculated as

$$HU = \sum_{h=1}^{24}(HUH) \tag{A3}$$

Four developmental stages are simulated (emergence, panicle initiation, flowering, and physiological maturity), the progression of which primarily depends on cumulative heat units which vary among cultivars.

Appendix A.1.2. Photosynthesis

Hourly leaf photosynthesis is based on an estimate of absorbed solar radiation and light use efficiency. This rate is calculated separately for a single shaded and sunlit leaf at each of three depths in the canopy using Equations (A4)–(A6):

$$A_n = A_m(1 - \exp\left(\frac{-\varepsilon I_a}{A_m}\right)) - R_d \tag{A4}$$

$$A_m = \left(\frac{49.57}{34.26}\right) \times \left(1 - \exp\left(-\frac{0.208(CO2 - 60)}{49.57}\right)\right) \tag{A5}$$

$$\varepsilon = \varepsilon 340\text{ppm} \frac{(1 - \exp(-0.00305 CO2 - 0.222))}{(1 - \exp(-0.00305 \times 340 - 0.222))} \tag{A6}$$

where $A_n$ is the net $CO_2$ assimilation rate (kg $CO_2$ ha$^{-1}$ leaf h$^{-1}$), $A_m$ is the $CO_2$ assimilation rate at light saturation, $\varepsilon$ is the initial light-use efficiency (kg $CO_2$ ha$^{-1}$ h$^{-1}$ (J m$^2$ s)$^{-1}$), and other variable and units were defined in Table A1. Note that $\varepsilon 340$ ppm is $\varepsilon$ at a $CO_2$ concentration of 340 ppm and linearly declines from a maximum of 0.54 when average temperatures rise above 10 °C as per [63]. In these equations, constant parameter values are used and assumed to be conserved across rice cultivars.

Leaf net photosynthesis is scaled up to the canopy level using a sunlit/shaded leaf approach. At each depth in the canopy, absorbed flux for diffuse, direct, and total PAR is computed for each leaf type using an exponential decay function for direct beam solar radar radiation as in Equation (A7) based on [64]:

$$I_a = (1 - \rho)I_0 \exp(-k \times L) \tag{A7}$$

where $I_a$ (J m$^{-2}$ ground s$^{-1}$) is the net photosynthetically active radiation (PAR) at depth $L$ in the canopy, $I_0$ (J m$^{-2}$ ground s$^{-1}$) is the photosynthetically active radiation at the top of the canopy, $L$ is cumulative LAI (m$^2$ leaf m$^{-2}$ ground), $\rho$ is the light reflection coefficient of the canopy, and $k$ is the canopy light extinction coefficient for PAR. Assimilation rate and absorbed radiation are integrated using a three-point Gaussian method at each canopy layer which is then scaled up to estimate the daily whole canopy net assimilation rate. Photosynthesis is allocated to the leaf, stem, and panicle with carbon partitioning coefficients that vary with plant phenology and water or nitrogen stress.

Appendix A.1.3. Temperature Stress

ORYZA includes the option for users to enter an observed spikelet number as an input value. If this option is used, heat sterility functions are not utilized, as this number is assumed to reflect the potential grain number. However, in the more typical case where spikelet number must be simulated in the model, the following methods are used to account for cold and hot temperature stresses.

Spikelet sterility is affected by cumulative daily average temperatures below 22 °C prior to anthesis as in Equations (8) and (9):

$$COLDTT = \sum_{p}^{f}(22 - T_d) \tag{A8}$$

$$S_c = 1 - \left(4.6 + 0.054 \times COLDTT^{1.56}\right)/100 \tag{A9}$$

where *COLDTT* is cumulative cold temperature degree-days (°Cd), $T_d$ is the daily average air temperature, $p$ and $f$ are the dates of panicle initiation and flowering, and $S_c$ is the percentage sterility caused by cold.

Spikelet fertility is also reduced when averaged $T_{max}$ at the flowering period is above a critical value as in Equation (A10), where the fraction of fertile rice spikelets reduced by heat ($S_h$) is estimated as:

$$S_h = 1/(1 + \exp(0.853(T_{max} - 36.6))) \tag{A10}$$

The actual stress factor on spikelet sterility at anthesis is calculated as the minimum of these two stresses and reduces the number of spikelets which are then associated with grain number and yield.

*Appendix A.2. Modified ORYZA Model*

Appendix A.2.1. Development

The original bilinear model was replaced by a beta function (Equation (A11)) to more accurately estimate hourly air temperatures. The normalized beta model ([65]) is given by the following equation:

$$HUH = \left\{ \left( \frac{T_h - T_{base}}{T_{opt} - T_{base}} \right) \left( \frac{T_{high} - T_h}{T_{high} - T_{opt}} \right)^{((T_{high} - h)/(T_{opt-base}))} \right\}^{TSEN} \tag{A11}$$

where *HUH*, $T_h$, $T_{base}$, $T_{opt}$ (°C), and $T_{high}$ were previously defined and TSEN determines the curvature of the response.

Appendix A.2.2. Coupled Gas Exchange Model

The three sub-models (FvCB, BWB versions, and energy balance) were interdependent and solved numerically using the Newton–Raphson method [66] through a nested iterative procedure developed by [67]. This modification simulated the direct effects of $CO_2$ on stomatal closure and subsequent increases in leaf temperature which in turn influenced photosynthesis and transpiration.

Leaf Photosynthesis

The Farquhar, von Caemmerer, and Berry (FvCB) [46] biochemical model of leaf photosynthesis as modified by [68] was used. The three rate-limiting steps in the FvCB model for net leaf photosynthesis ($A_n$) were the ribulose 1·5-bisphosphate carboxylase/oxygenase (Rubisco) rate ($A_c$), the ribulose-1,5-bisphosphate (RuBP) regeneration/electron transport-limited rate ($A_j$), and the triose phosphate utilization ($T_p$)-limited rate ($A_P$) (Equation (A12) through Equation (A15)). Other inputs included the intercellular $CO_2$ partial pressure ($C_i$) estimated as a fraction of ambient $CO_2$ partial pressure.

$$A_n = \min\left( A_c, A_j, A_P \right) \tag{A12}$$

$$A_c = \frac{(C_i - \Gamma_*) V_{cmax}}{C_i + K_c \left( 1 + \frac{O}{K_o} \right)} - R_d \tag{A13}$$

$$A_j = \frac{(C_i - \Gamma_*) J}{4(C_i + 2\Gamma_*)} - R_d \tag{A14}$$

$$A_P = 3T_p \tag{A15}$$

$$C_i = C_a \times 0.7 \tag{A16}$$

$$J = \frac{\sigma I_a + J_{max} - \sqrt{(\sigma I_a + J_{max})^2 - 4\theta I_a J_{max}}}{2\theta} \tag{A17}$$

Parameter values used in this study were obtained from [33]. Temperature response for $V_{cmax}$, $J_{max}$ and $T_p$ were expressed in Equation (A18), where *c* is a scaling constant, $\Delta H_a$ is the enthalpy of activation, $T_L$ is leaf temperature in °C and *R* is the universal gas constant (8.314 J mol$^{-1}$ K$^{-1}$), and *parameter*$_{25}$ is the value of a parameter at 25 °C, $\Delta H_d$ is the enthalpy of deactivation and $\Delta S$ is entropy. An estimate of each parameter at 25 °C was obtained using the method of [69].

$$T_p = parameter_{25} \times \frac{e^{\left( c - \frac{\Delta H_a}{R \times (273.15 + T_L)} \right)}}{1 + e^{\left( \Delta S \times (273.15 + T_L) - \frac{\Delta H_d}{R \times (273.15 + T_L)} \right)}} \tag{A18}$$

Stomatal Conductance

The Ball–Woodrow–Berry model (BWB) was used to simulate leaf stomatal conductance to water vapor [47]:

$$g_s = g_0 + g_1 A_n \frac{h_s}{C_s} \tag{A19}$$

where $g_s$ is stomatal conductance ($\mu$mol m$^{-2}$ s$^{-1}$), $h_s$ is humidity at the leaf surface, $C_s$ is leaf surface $CO_2$ concentration, $g_0$ is the minimum conductance (mol m$^{-2}$ s$^{-1}$), and $g_1$ is a slope parameter. Parameter values used in this study were time invariant with $g_0$ at 0.10 and $g_1$ at 8.20.

Leaf Energy Balance Model

Leaf temperature ($T_L$) was determined by a linear solution of the leaf energy budget [70]:

$$T_L = T_h + \frac{R_{abs} - \varepsilon \sigma T_h^4 - \lambda g_v VPD/P_a}{c_p(g_h + g_r) + \lambda \left( \left( \frac{de_s(T_h)}{dT} \right)/P_a \right) g_v} \tag{A20}$$

where $T_h$ (°C) is hourly air temperature, $R_{abs}$ (W m$^{-2}$) is absorbed long wave and short-wave radiation per surface leaf area, $\varepsilon$ (0.97) is leaf thermal emissivity, $\sigma$ (5.67 × 10$^{-8}$, W m$^{-2}$ K$^{-4}$) is Stefan–Boltzman constant per surface area, $\lambda$ (44, kJ mol$^{-1}$) is the latent heat of vaporization at 25 °C, $g_v$ (mol m$^{-2}$ s$^{-1}$) is total water vapor conductance per surface leaf area, $VPD$ (kP$_a$) is vapor pressure deficit of ambient air, $P_a$ (kP$_a$) is atmospheric pressure, $c_p$ (J mol$^{-1}$ C$^{-1}$) is specific heat of air, $g_h$ (mol m$^{-2}$ s$^{-1}$) is heat conductance for boundary layer per surface leaf area, $g_r$ (mol m$^{-2}$ s$^{-1}$) is radiative conductance per surface leaf area, and $e_s$ (kP$_a$) is vapor pressure at leaf surface, and $g_b$ (mol m$^{-2}$ s$^{-1}$) is boundary layer conductance to water vapor.

Root mean square error and Wilmott's index of agreement [51] were given in Equations (A21) and (A22).

$$RMSE = \sqrt{N^{-1} \sum_{i=1}^{N} (O_i - S_i)^2} \tag{A21}$$

$$d = 1 - \frac{\sum_{i=1}^{N} (S_i - O_i)^2}{\sum_{i=1}^{N} \left( |S_i - \overline{O}| + |O_i - \overline{O}| \right)^2}, \quad 0 \le d \le 1 \tag{A22}$$

where, $N$ is the number of observations, and $O_i$ and $S_i$ are observed and simulated values for observation $I$, and $d$ is the agreement index value.

**Table A1.** Equation variables and units.

| Variable | Description | Unit | Equation Number |
|---|---|---|---|
| $A_n$ | canopy or leaf net photosynthetic rate | $\mu$mol m$^{-2}$ s$^{-1}$ | (A4), (A12) and (A19) |
| $I_a$ | photosynthetically active radiation incident to leaf surface | $\mu$mol m$^{-2}$ s$^{-1}$ | (A4), (A7) |
| $\varepsilon$ | light use efficiency | kg $CO_2$ ha$^{-1}$ h$^{-1}$ (J m$^2$ s$^{-1}$) | (A4), (A6) |
| $A_m$ | $CO_2$ assimilation rate at light saturation | kg $CO_2$ ha$^{-1}$ leaf h$^{-1}$ | (A4), (A5) |
| $R_d$ | day respiration | $\mu$mol m$^{-2}$ s$^{-1}$ | (A4), (A13) and (A14) |
| $I_0$ | photosynthetically active radiation at the top of the canopy | J m$^{-2}$ ground s$^{-1}$ | (A7) |
| $L$ | cumulative LAI | m$^2$ leaf m$^{-2}$ ground | (A7) |
| $\rho$ | light reflection coefficient of the canopy | | (A7) |
| $k$ | canopy light extinction coefficient for PAR | | (A7) |
| $A_c$ | Rubisco carboxylation-limited rate | $\mu$mol m$^{-2}$ s$^{-1}$ | (A12), (A13) |
| $A_P$ | triose phosphate utilization ($T_p$)-limited photosynthetic rate | $\mu$mol m$^{-2}$ s$^{-1}$ | (A12), (A15) |
| $A_j$ | RuBP regeneration-or electron transport-limited rate | $\mu$mol m$^{-2}$ s$^{-1}$ | (A12), (A14) |
| $C_i$ | Intercellular $CO_2$ concentration | $\mu$bar | (A13), (A16) |

**Table A1.** *Cont.*

| Variable | Description | Unit | Equation Number |
|---|---|---|---|
| $\Gamma^*$ | the $CO_2$ compensation point in the absence of $R_d$ | $\mu mol\ m^{-2}\ s^{-1}$ | (A13) |
| $V_{cmax}$ | maximum carboxylation rate | $\mu mol\ m^{-2}\ s^{-1}$ | (A13) |
| $Kc$ | Michaelis constant of Rubisco affinity for carbon dioxide | $kP_a$ | (A13) |
| $O$ | partial pressure of oxygen at Rubisco | $kP_a$ | (A13) |
| $Ko$ | Michaelis constant of Rubisco affinity for carbon dioxide | $kP_a$ | (A13) |
| $J$ | photosystem (PS) II electron transport rate | $\mu mol\ m^{-2}\ s^{-1}$ | (A17) |
| $T_P$: | triose phosphate utilization | $\mu mol\ m^{-2}\ s^{-1}$ | (A15), (A18) |
| $C_a$ | ambient $CO_2$ concentration | $\mu bar$ | (A16) |
| $J_{max}$ | potential maximum electron transport rate | $\mu mol\ m^{-2}\ s^{-1}$ | (A17) |
| $\sigma$ | electron transport efficiency of PS II | – | (A17) |
| $\theta$ | a curvature parameter | – | (A17) |

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
