# Peer review of "Improving Simulations of Rice in Response to Temperature and CO2"

_agronomy, doi:10.3390/agronomy12122927_

Round 1

Reviewer 1 Report

 This manuscript reports on the evaluation of modifications on the rice model ORYZA to simulate the interaction effects of Temperature and CO2 increase on rice crop biomass production and yield. The authors approach is well described and the results suggest that improving representation of high T effect on yield is more determinant to capture the effect of the interaction of T and CO2 increase   and changes in using formalism for CO2 representation for a more mechanistic approach did not make significant difference.

 The manuscript  is well structured and easy to read. Conclusion is supported by the results. The main finding confirmed the limitation of the model ORYZA to account for high Temperature stress which may not be novel  but is informative on the value of alternative for the model improvement  to account for Temperature  estimate within the flowering time window and on the need of other options to improve CO2 assimilation representation.

Author Response

No points to respond to from reviewer. We have gone through the entire manuscript and spell-checked and cleaned up grammar throughout the revision.

Reviewer 2 Report

1.    The references in the paper are not new publication, too old, more than half of which are articles five years ago, except for a few self-citations is new. It is suggested that the author add references in recent years and update the Introduction. The IPCC has released the latest report, which can be used as your reference.

2.    Radiation is a key input variable in every crop models. I hope you can discuss this variable in your paper. In addition, could you explain how water and nitrogen are considered in your experiment.

3.    Can you simplify Appendix A? This part is not the main body of the research article. It doesn't need to be so much and complicated.

Author Response

  1. The references in the paper are not new publication, too old, more than half of which are articles five years ago, except for a few self-citations is new. It is suggested that the author add references in recent years and update the Introduction. The IPCC has released the latest report, which can be used as your reference.

Five of the references in introduction were updated (4, 8-11) with recent publications and the latest IPCC report. Some of the older references were also removed for conciseness. 

  1. Radiation is a key input variable in every crop models. I hope you can discuss this variable in your paper. In addition, could you explain how water and nitrogen are considered in your experiment?

      We added a new paragraph in the end of the Discussion section, and a new Table S3, that showed correlations between cumulative seasonal solar radiation and yield.  Correlations were higher for field grown cultivars presumably because the variation in planting dates as a treatment factor in that dataset provided for a wider range of solar radiation data for which to test the relationship.  In contract, this association was weaker in the controlled environment studies, probably because most chambers were exposed to the same solar radiation amount and heat stress was more prevalent at anthesis. Just as interesting, we noted that model versions which contained the revised heat stress methods and gas exchange approach exhibited stronger correlations with solar radiation than the original ORYZA, suggesting that these versions were able to capture more of the agronomic relations more intuitively.

Rice was simulated without limitation of irrigation and fertilization since at the experimental sites rice was grown under sufficient irrigation and fertilization. We added this to the last paragraph in the M&M section.

 3.     Can you simplify Appendix A? This part is not the main body of the research article. It doesn't need to be so much and complicated.

In prior reviews from an earlier submission, we were asked to provide more detail!  To balance, we removed multiple extraneous variable descriptions that had previously been defined in the equations.   We also shorted a few sentences in the M&M section which described the modeling algorithms in too much detail.

In general, we also revised the Discussion to be more focused as well to strengthen the Conclusion section.

Reviewer 3 Report

The manuscript titled ‘Improving Simulations of Rice in Response to Temperature and CO2’ investigated how widely used ORYZA rice model’s extension with a revised heat stress component, and a coupled leaf-level gas exchange algorithm effect yield and spikelet fertility predictions.

The authors in their previous publications (eg.: 20 and 30 in references) worked on similar databases, but new modeling procedures are being tested and developed by comparison with measured data. The modeling environment mainly spans to higher CO2 and temperature regimes than optimal, as these probably be of particular importance for assessing climate impacts more accurately.

Their results emphasize the need to improve carbon assimilation methodology and to focus on assessing the impact of extreme temperature during critical developmental stages. They also highlighted the need to consider heat sensitivity differences among cultivars.

The research covers an important topic thus the information provided would be in the interest of the Agronomy journal’s readership.

However, the appearance of the publication needs improvement, minor comments should be addressed before publication.

Line 260. Fig. 1. a and b are edited differently. Please unify the appearance of the figure. There is also an error in the unit of measurement. The appearance of the figures would be more uniform if the appearance were harmonized with Fig. 3. as well. Please avoid using gray axis labels instead of black ones.

Line 38. and 57. It seems that there is a correction/modification left in it.

Line 105. I don't understand the marking of references. Why not use the designation [31-35]?

Line 443. Are you sure the plant density was 36 plants m-2? Compared to 236, it seems small.

Line 294. Fig 2. would look better if the x-axis extended beyond the last data.

Line 324. In Table 5 in the second line, the marking has slipped, please correct it.

Line 456. Table S1. In my opinion, it would be good to harmonize the colors of the markers used in figures a and b in the case of ‘susceptible’ data points.

Line 609. Table A1. The empty part left free under Vcmax seems unreasonable to me, please correct it.

Author Response

Line 260. Fig. 1. a and b are edited differently. Please unify the appearance of the figure. There is also an error in the unit of measurement. The appearance of the figures would be more uniform if the appearance were harmonized with Fig. 3. as well. Please avoid using gray axis labels instead of black ones.

All figures were updated as suggested

Line 38. and 57. It seems that there is a correction/modification left in it.

Correction in Line 38 and 57 was accepted.

Line 105. I don't understand the marking of references. Why not use the designation [31-35]?

This was a typo mistake and we have revised it as suggested.

Line 443. Are you sure the plant density was 36 plants m-2? Compared to 236, it seems small.

We verified these numbers with the original articles from which we obtained this source data. Different experimental units were used by those authors of the experiments. For example, the IR30 cultivar was grown in larger soilbins with 2m2 production area and directly sown into a soil media.  The other varieties were grown in pots in smaller units.  We added additional footnote of explanation to the Table S1.

Line 294. Fig 2. would look better if the x-axis extended beyond the last data.

The x-axis of figure 2 was extended.

Line 324. In Table 5 in the second line, the marking has slipped, please correct it.

Corrected as suggestion and standardized for other tables.

Line 456. Table S1. In my opinion, it would be good to harmonize the colors of the markers used in figures a and b in the case of ‘susceptible’ data points.

We have revised as suggested for Figure S1 (not Table S1). Thank you so much for the suggestion. 

Line 609. Table A1. The empty part left free under Vcmax seems unreasonable to me, please correct it.

Corrected as suggested.

Reviewer 4 Report

This simulation study evaluated ORYZA modifications of heat stress and leaf gas exchange algorithm using observed rice yields under different temperature and CO2 concentration conditions. The conclusion were generally supported by results, however several issues are needed to be addressed: (1) the effect of temperature increase on rice development should be discussed for the different model versions which should influence rice yield formation greatly; (2) since the modifications included leaf energy balance simulation, the leaf temperature could be different from the original version (if it does not include the leaf energy balance module), the model improvements in simulating rice yield could be contributed by both better simulated leaf temperature and better rice response to temperature stress, which should be discussed; (3) the discussion part should be shortened and focused on the model comparisons in the current study (such as which modification is more important for simulating rice yield in response to temperature and CO2 levels), not only addressing the future works; (4) many long sentence is difficult to be understood, and it will be better to rewrite them into short sentences.

Specific comments:

L20 RMSE should be defined when it was first used.

L191-L193 should be moved to the previous part.

L231-L231: any simulated data to support this statement?

L411: any measurement to compare the simulated LAI to support this hypothesis?

Fig. 1 and Fig. 2: Is any measured errors available in these figures? Some over or under simulations may be not statistically significant considering the measured errors. For Fig. 2, it looks the original version showed better response to CO2 levels than other two versions, but with consistent under-simulations compared with observed yield.

Author Response

(1) This simulation study evaluated ORYZA modifications of heat stress and leaf gas exchange algorithm using observed rice yields under different temperature and CO2 concentration conditions. The conclusion were generally supported by results, however several issues are needed to be addressed: (1) the effect of temperature increase on rice development should be discussed for the different model versions which should influence rice yield formation greatly;

We included a new Table S3 and a new paragraph in the Discussion section regarding the relationship with temperature and yields. Specifically we indicated via linear regression that correlation between simulated yield and different growing duration stages (e.g. emergence to flowering;  flowering to maturity) existed and illustrate how the different model versions varied in terms of the strength of the temperature response. Just as interesting, we documented how the newer model versions consistently exhibited higher correlations with temperature and solar radiation which would conform more closely with our intuitions regarding these types of agronomic relationships.

(2) since the modifications included leaf energy balance simulation, the leaf temperature could be different from the original version (if it does not include the leaf energy balance module), the model improvements in simulating rice yield could be contributed by both better simulated leaf temperature and better rice response to temperature stress, which should be discussed;

We included this several lines about this in the revised discussion. As simulated, canopy / leaf temperatures can be higher under elevated versus ambient CO2 with the new methodology (up to about 0.4 to 0.5). This may have improved some of the response to CO2  x T treatments. However, this response did not necessarily improve the heat stress results over the original model. This was because the model still uses air temperature for developmental responses while canopy temps influence leaf level gas exchange.  We summarized these lines in the new Discussion:

Previous work with this approach showed how the methodology can simulate reduced stomatal conductance in response to rising CO2 concentration. This in turn decreases transpiration, increases leaf and canopy temperature (as much as 0.5°C, not shown) and exerts an additional influence on photosynthesis [39]. However, despite these improvements, ORYZA-V3 model over-estimated vegetative biomass for some SPAR datasets (Table 1), which may reflect the need to also account for photosynthetic acclimation [31], substrate induced feedback inhibition [64, 65], and/or the influence of nitrogen and light attenuation in the canopy on photosynthetic properties [37]

 (3) the discussion part should be shortened and focused on the model comparisons in the current study (such as which modification is more important for simulating rice yield in response to temperature and CO2 levels), not only addressing the future works;

We merged several of the ideas in the original version of the Discussion to substantially improve brevity and indicate the improvements in model versions with respect to T and CO2 simulations. We also added a paragraph that provided an additional comparison with model versions in response to temperature and solar radiation (new lines 380-392).

(4) many long sentence is difficult to be understood, and it will be better to rewrite them into short sentences.

Agreed. Significant effort was made by native-english speaking co-authors to address grammatical mistakes and shorten sentences throughout the entire manuscript. Redundancy was removed as well where possible.

Specific comments:

L20 RMSE should be defined when it was first used.

RMSE (Root Mean Square Error) is now defined in L20

L191-L193 should be moved to the previous part.

L191-l193 was moved to section 2.2

L231-L231: any simulated data to support this statement?

We included the rational in the Discussion section in response to this comment and added an additional reference for why this may have occurred.  Observed leaf area was not available to confirm simulations with experimental data, so the rationale is based on concept instead of empirical proof at this point (new lines 393 – 399):

All model versions over-estimated aboveground biomass at air temperatures above 36°C (Table 1). This was likely due to over-predictions of leaf area index as reported by [27], who suggested modelers consider reduced assimilate partitioning to the leaves at lower or higher temperatures. [69] also observed differences in the ratio of whole plant leaf area to dry weight when grown under different temperatures. This suggests spe-cific leaf area values may need to be modified based on growth temperature to properly account for leaf area expansion and associated carbohydrate requirements.

L411: any measurement to compare the simulated LAI to support this hypothesis?

We include this rational as indicated above, but also added an additional supporting literature for our statement.  Experimental data unfortunately was not available to confirm. See comments above.

Fig. 1 and Fig. 2: Is any measured errors available in these figures? Some over or under simulations may be not statistically significant considering the measured errors. For Fig. 2, it looks the original version showed better response to CO2 levels than other two versions, but with consistent under-simulations compared with observed yield.

Agreed.  We included observed data error bars as obtained from the experimental dataset publication for Figure 1. Observed error bar for Figure 2 was not available since only average yield values were provided from the literature source for that study. We only see the original model response to CO2 to be better than the V3 version for the 160 ppm concentration which is not a realistic real world value. In this case, we indicated in the text that V3 was more accurate over the broader range of CO2 response and state V3 performed less well than V1 for sub-ambient CO2. (new lines 271-275).

Round 2

Reviewer 4 Report

I am satisfied to the authors' modifications on the manuscript which was impoved significantly.

I only have some minor comments now:

L91-94: These stataments were related to the method part closly, and it will be better to be moved to the Meterial and method part.

L185: 2.4 should be 2.3; Model calibration and simualtions can be changed as Model calibration and evaluation

 L212: 3.1.  SPAR can be changed to SPAR chamber data as comparing with the 3.2 Evaluation with Field Data

Author Response

All minor revisions were incorporated as shown with 'track changes' on the submitted revised manuscript.  Again, we much appreciate these critiques to improve our manuscript.